# T Cell Receptor-Directed Bispecific T Cell Engager Targeting MHC-Linked NY-ESO-1 for Tumor Immunotherapy

**DOI:** 10.3390/biomedicines12040776

**Published:** 2024-04-01

**Authors:** Yiming Li, Wenbin Zhao, Ying Shen, Yingchun Xu, Shuqing Chen, Liqiang Pan

**Affiliations:** 1Institute of Drug Metabolism and Pharmaceutical Analysis, College of Pharmaceutical Sciences, Zhejiang University, Hangzhou 310058, China; yiming_li@zju.edu.cn (Y.L.); pharmacy_zwb@zju.edu.cn (W.Z.); shenying925@zju.edu.cn (Y.S.); ycxu66@163.com (Y.X.); 2Zhejiang University Innovation Institute for Artificial Intelligence in Medicine, Engineering Research Center of Innovative Anticancer Drugs, Ministry of Education, Hangzhou 310018, China

**Keywords:** immunotherapy, bispecific T cell engager, T cell receptor, pMHC, NY-ESO-1, soluble expression

## Abstract

Antibody-based bispecific T cell engagers (TCEs) that redirect T cells to kill tumor cells have shown a promising therapeutic effect on hematologic malignancies. However, tumor-specific targeting is still a challenge for TCEs, impeding the development of TCEs for solid tumor therapy. The major histocompatibility complex (MHC) presents almost all intracellular peptides (including tumor-specific peptides) on the cell surface to be scanned by the TCR on T cells. With the premise of choosing optimal peptides, the final complex peptide–MHC could be the tumor-specific target for TCEs. Here, a novel TCR-directed format of a TCE targeting peptide–MHC was designed named IgG-T-TCE, which was modified from the IgG backbone and prepared in a mammalian cell expression system. The recombinant IgG-T-TCE-NY targeting NY-ESO-1_157–165_/HLA-A*02:01 could be generated in HEK293 cells with a glycosylated TCR and showed potency in T cell activation and redirecting T cells to specifically kill target tumor cells. We also found that the in vitro activity of IgG-T-TCE-NY could be leveraged by various anti-CD3 antibodies and Fc silencing. The IgG-T-TCE-NY efficiently inhibited tumor growth in a tumor–PBMC co-engrafted mouse model without any obvious toxicities.

## 1. Introduction

The T cell engager (TCE), which redirects cytotoxic T cells to kill tumor cells, is emerging as a promising strategy to treat cancer patients, especially those with a low-mutation-burden tumor or a low number of tumor-specific T cells who do not respond to checkpoint inhibitors [1]. In general, the TCE is comprised of two domains: one targets the tumor antigen on the tumor cell surface and the other binds the protein (typically the CD3 delta) on the T cell membrane. Over decades, there has been much effort directed toward the design of TCE formats based on the variable domain of antibodies. To date, more than 20 platforms have made their way into clinical development [2], and eight TCEs have been approved by the FDA. Although the efficacy of these drugs is impressive, most TCEs cause harmful ‘on-target off-tumor’ effects due to the unfavorable specificity of tumor-associated antigen (also expressed on normal cells) on the tumor cells [3]. On the other hand, it is challenging to seek tumor-specific antigens, as the target antigens with an extracellular domain that is accessible to antibodies comprise only <10% of the proteome [4], and the mutations that differentiate tumor cells from normal cells may be buried inside of the protein.

Major histocompatibility complex (MHC) can present peptides derived from intracellular proteins (>90% of the proteome) to the cell surface to form peptide–MHC (pMHC), which is recognized by the T cell receptor (TCR) on T cells. The pMHCs extend the range of antigens on the membrane and are regarded as tumor-specific antigens when the peptide is tumor-specific (usually mutated or with highly restricted expression in normal tissues) [5]. However, employing TCR as a soluble drug to target pMHC is challenging due to the poor water solubility of TCR [6]. Over decades, several protein-engineering solutions have been developed to generate stable and soluble TCRs, including stabilizing mutations of the TCR C domain, fusing a soluble protein part to the TCR C terminus, and other bespoke stabilizations of individual TCRs [6]. The first reported TCR-based TCE was ImmTACm which comprises an affinity-matured TCR fused to a humanized CD3-specific single-chain antibody fragment (scFv) [7] and exhibits high efficiency in killing tumor cells (EC_50_ at picomole level). One ImmTAC molecule named Tebentafusp has been approved by the FDA for treating uveal melanoma [8]. However, the half-life of ImmTAC is only several hours, which requires more frequent dosing [7], and it is not straightforward to produce ImmTAC with the correct structure due to the complex refolding procedure. Therefore, more formats of T-TCE with distinct properties need to be explored.

The crystalline fragment (Fc) of an antibody is a widely used part for fusing to improve the half-life and water solubility of a protein. Recently, soluble expression of T-TCE with Fc in mammalian cells was reported by Froning et al. with a simple method [9]. Inspired by this, our team previously reported a soluble click-linked T-TCE but the killing efficiency of T-TCE in single uses was not satisfactory [10]. Herein, we describe a new IgG-like T-TCE (IgG-T-TCE) format with the tumor antigen-targeting arm (VH and VL) of IgG replaced by an evolved TCR. The IgG-T-TCE molecules were expressed in mammalian cells following simple purification (Ni column and anti-Flag column). For evaluating the format, we set a well-studied paired TCR/pMHC as a representative: 1G4-113 TCR and NY-ESO-1_157–165_/HLA-A*02:01 (NY-ESO-1, a cancer–testis antigen; HLA*A-02:01, one of the most common MHC alleles in human) [11]. The purified product showed retained affinities to the tumor target and human CD3 and the ability to activate T cells and specifically kill target tumor cells in vitro. We also evaluated the impact of various anti-CD3 antibodies and Fc silencing on the in vitro activities of the IgG-T-TCE-NY. Finally, the IgG-T-TCE-NY efficiently inhibited tumor growth in a tumor–PBMC co-engrafted mouse model without any obvious toxicities, demonstrating its potential for precise tumor immunotherapy in the future.

## 2. Materials and Methods

### 2.1. Cells and Culture Conditions

Human embryonic kidney 293F cells (HEK293F) were kindly provided by the Comprehensive AIDS Research Center (Tsinghua University, Beijing, China) and rotary-cultured in an SMM 293-TI medium (Sino Biological Inc., Beijing, China).

Cell lines A375 (melanoma), Jurkat (T cell leukemia), and K562 (chronic myelogenous leukemia) were purchased from American Type Culture Collection (ATCC, Manassas, VA, USA). Cell line U266 (multiple myeloma) was purchased from BeNa Culture Collection (BNCC, Beijing, China).

The construction of the A375-NY-GFP, K562-NY (NY-ESO-1_157–165_^+^, HLA-A*02:01^+^, GFP^+^) and K562-Ctrl (irrelevant peptide, HLA-A*02:01^+^, GFP^+^) cell lines has been described previously [10].

All tumor cells here were maintained in RPMI-1640 (Gibco, Grand Island, NY, USA) with 10% fetal bovine serum (FBS; Gibco, Grand Island, NY, USA) at 37 °C with 5% CO_2_ and 95% humidity.

Human peripheral blood mononuclear cells (PBMCs) were purchased from Shanghai Saily Biotechnology Co., Ltd. (Shanghai, China). Generally, PBMCs were allowed to rest in RPMI-1640 with 10% FBS for several hours before use.

### 2.2. Design, Construction, Expression, and Purification of IgG-T-TCE-NY

The design of the IgG-like T cell engager was based on the backbone of conventional antibody IgG1 (Figure 1a). Briefly, two binding domains targeting pMHC and human CD3 were located at two short arms in a TCR-fused format and a Fab, respectively. The TCR consisted of the variable domain and partial constant domain that was extracellular. The constant domains of the TCR and antibody were linked by a GS linker and a pair of cysteine mutations (T48C in TRAC, S57C in TRBC2) was inserted to improve the stability of the TCR. For reducing mis-paired products, the ‘knob into hole’ strategy for Fc and a CrossMAb design were applied. Moreover, two light chains were fused with the His tag and Flag tag, respectively, to improve the purity of the purified products. 1G4-113 TCR targeting NY-ESO-1_157–165_/HLA-A*02:01 [12] was set as a representative TCR and a humanized OKT3 clone was chosen to bind human CD3. The genes encoding four chains of IgG-T-TCE-NY were cloned into vector pLVX-Puro and transiently transfected into HEK293F cells using linear polyethyleneimine (PEI; Polysciences, Warrington, PA, USA) together.

The supernatant of HEK293F was collected after 3–4 days’ culture and filtered through a 0.45 μm filter unit, followed by two-step purification using a Ni column (HisTrapTM HP; GE Healthcare Life Sciences, Little Chalfont, Buckinghamshire, UK) and Anti-Flag M2 Affinity Gel (Millipore Sigma, St. Louis, MO, USA). The purified product was then concentrated in phosphate buffered saline (PBS), aliquoted, and kept at −80 °C for long-term storage.

As for the variants of IgG-T-TCE-NY, Fc was silenced by LALAPG substitutions and OKT3 was replaced with OKT3-LT (lower affinity to CD3 [13]) or UCHT1 (higher affinity to CD3 [14]).

### 2.3. Characterization of IgG-T-TCE-NY by SDS-PAGE, WB analysis, ELISA, and Flow Cytometry

SDS-PAGE and WB analysis were performed to identify the components of the purified product with a common operation. As a note, deglycosylation was conducted using the Enzymatic In-Solution N-Deglycosylation Kit (Sigma, St. Louis, MO, USA) overnight before reducing SDS-PAGE; the chains with a His tag or Flag tag were labeled with His Tag (C-terminal Specific) Mouse Monoclonal Antibody (Beyotime, Shanghai, China) or Flag Tag Mouse Monoclonal Antibody (Beyotime, Shanghai, China).

The affinity of IgG-T-TCE-NY to pMHC was tested via ELISA as previously described [10]. Briefly, a 96-well EIA/RIA plate was coated with streptavidin (2 µg/mL) overnight at 4 °C. After blocking, biotin-labeled pMHCs (2 µg/mL, generated by refolding as in a previous report [15]) were added for 2 h at 37 °C. Then, IgG-T-TCE-NY and its controls were added for 1 h at 37 °C, followed by incubation with HRP-conjugated goat anti-human IgG (H + L) (1:1000 dilution, Beyotime, Shanghai, China) for 1 h at 37 °C. The samples finally reacted with the TMB substrate solution and the reaction was stopped by adding 2M H_2_SO_4_. The absorbance of samples at 450 nm was measured using a Model 680 Microplate Reader (Bio-Rad, Hercules, CA, USA). Washing with PBST was needed between each step.

The affinity of IgG-T-TCE-NY to human CD3 was tested via flow cytometry as previously described [10]. Jurkat cells (expressing human CD3) were used as target cells. Briefly, target cells were harvested and incubated with serial dilutions of IgG-T-TCE-NY or controls for 30 min at 4 °C. Then, cells were incubated with FITC-conjugated goat anti-human IgG (H + L) (1:200 dilution, Beyotime, Shanghai, China) for 30 min at 4 °C. Samples were analyzed with an ACEA NovoCyte^TM^ flow cytometer (ACEA Biosciences, San Diego, CA, USA). Washing with PBS was needed between each step. In the experiment testing simultaneous binding of IgG-T-TCE-NY to NY-ESO-1_157–165_/HLA-A*02:01 and human CD3, similar operations were executed, except incubation with a biotin-labeled pMHC and SA-PE was used instead of FITC-conjugated goat anti-human IgG following the IgG-T-TCE-NY incubation.

### 2.4. In Vitro Cytotoxicity Assay

The cytotoxicity of tri-specific T cell engagers was evaluated using apoptosis detection or the LDH-releasing assay. Human PBMCs and target tumor cells were incubated in a 4:1 ratio in the presence of serial dilutions of tri-specific T cell engagers or controls in both experiments. For apoptosis (constructed cell lines that carried GFP), mixed cells were generally incubated in 48-well plates in RPMI-1640 with 10% FBS. After 2 days, samples were harvested, stained using an Annexin V 633 Apoptosis Detection Kit (Dojindo, Kumamoto, Japan), and analyzed with an ACEA NovoCyte^TM^ flow cytometer. For LDH, mixed cells were generally incubated in 96-well plates in RPMI-1640 (without phenol red) with 1.5% FBS. Additional control wells were set up with PBMCs alone, tumor cells alone, or medium alone to calculate the spontaneous LDH value from cells. After 2–3 days, the supernatants were transferred into new 96-well plates. Following the steps of the LDH Cytotoxicity Assay Kit (Beyotime, Shanghai, China), the released LDH was represented by the absorbance at 490 nm. The lysis percentage was calculated as (experimental well − spontaneous PBMCs − spontaneous tumor cells − medium alone)/(MAX tumor cells alone − tumor cells alone) × 100. Three repeats were used for each sample.

### 2.5. T Cell Activation, Proliferation, and Cytokine Analysis

The early activation marker CD69 and late activation marker CD25 on the surface were evaluated using flow cytometry to confirm the activation state of T cells. Briefly, cells were harvested after 2 days of co-culture and stained with the following antibodies before detection: Super Bright 436 anti-human CD69 (Invitrogen, Carlsbad, CA, USA), PE anti-human CD25 (Invitrogen, Carlsbad, CA, USA), and APC anti-human CD3 (Invitrogen), APC anti-human CD8 (Invitrogen, Carlsbad, CA, USA), or APC anti-human CD4 (Invitrogen, Carlsbad, CA, USA).

For T cell proliferation, PBMCs were labeled with CFSE (Invitrogen, Carlsbad, CA, USA) according to the manufacturer’s protocol before co-culture. After 3–8 days of co-culture, cells were harvested, stained with PI (Invitrogen, Carlsbad, CA, USA), and analyzed with flow cytometry.

Cytokine IL-2 and Granzyme B were detected using ELISA and flow cytometry, respectively. Cells and the supernatants were collected after 2–3 days of co-culture. IL-2 in the supernatant was measured according to the Human IL-2 Precoated ELISA kit (Dakewe, Shenzhen, China). As for intracellular Granzyme B, cells were fixed and permeabilized using an eBioscience™ Intracellular Fixation and Permeabilization Buffer Set (Invitrogen, Carlsbad, CA, USA) as indicated by the manufacturer before detection.

### 2.6. In Vivo Antitumor Activity

Fifteen eight-week-old female NOD-SCID mice (GemPharmatech, Nanjing, China) were randomly divided into three groups and given different treatments: no PBMC group—A375 (s.c.) and PBS (i.v.); vehicle control group—PBMC + A375 (s.c.) and PBS (i.v.); IgG-T-TCE group—PBMC + A375 (s.c.) and IgG-T-TCE-NY (i.v.). In detail, mice were subcutaneously engrafted with 1.5 × 10^6^ tumor cells and 3.75 × 10^6^ PBMCs together or tumor cells alone in the right armpit on day 0. Each group was administrated intravenously with 105 µg/kg IgG-T-TCE-NY or PBS on day 0, day 3, and day 6. The tumor volume was monitored and recorded according to the following formula: volume = (length × width^2^)/2. Mice were euthanized when there was a mouse with a tumor larger than 800 mm^3^ and tumors were taken from mice. The animal work was performed following the protocol approved by the Committee on the Ethics of Animal Experiments of Zhejiang University (Hangzhou, China, 17526).

## 3. Results

### 3.1. Soluble Expression and Characterization of IgG-T-TCE

The designed IgG-like T-TCE was transiently expressed by HEK293F cells and purified through the Ni column and anti-Flag column to reduce mis-paired products (Figure 1b). IgG-T-TCE-NY was set as a representative composed of a TCR targeting NY-ESO-1_157–165_/HLA-A*02:01 (1G4-113 TCR) and an anti-human CD3 variable domain from the antibody (humanized OKT3). Purified protein was identified by reducing SDS-PAGE but the bands did not match the theoretic MW well (Figure 1d). To make sure of the positions of the A chain and L chain, Western blot analysis for the His tag and Flag tag was undertaken, and the results showed a correct position for the L chain but an upper position for the A chain (Figure 1c), which was consistent with the result of the SDS-PAGE. Considering that TCR’s N-glycosylation is common in mammalian cells, we deglycosylated the purified product by adding PNGase F before the reducing SDS-PAGE. The deglycosylated sample appeared with clear bands sitting on proper positions (Figure 1d).

We then evaluated the affinity of IgG-T-TCE-NY to NY-ESO-1_157–165_/HLA-A*02:01 and human CD3 with ELISA using the NY-ESO-1_157–165_/HLA-A*02:01 monomer (Appendix A) and with FACS using the Jurkat cell line, respectively. In the ELISA assay (Figure 1e), the IgG-T-TCE group showed a positive result compared with the blank group (not added IgG-T-TCE-NY), indicating IgG-T-TCE-NY retained the ability of TCR to bind NY-ESO-1_157–165_/HLA-A*02:01. The glycosylation on TCR did not reduce the binding of TCR with target pMHC; instead, the binding was slightly stronger, which may be explained by glycan stabilizing TCR (Figure 1f). The result of FACS showed increased MFI with increased concentration of IgG-T-TCE-NY (Figure 1g), indicating the ability of IgG-T-TCE-NY to bind human CD3. Furthermore, IgG-T-TCE-NY simultaneously binding NY-ESO-1_157–165_/HLA-A*02:01 and human CD3 was identified using a sandwich method (Jurkat + IgG-T-TCE-NY + NY-ESO-1_157–165_/HLA-A*02:01-bio + streptavidin-PE) with FACS (Figure 1h).

### 3.2. IgG-T-TCE Induced T Cell Activation

T cell activation was the pre-step for T cell engagers to kill tumor cells. T cells in an activated state change in several respects compared with naive T cells, such as surface protein expression, cytokine secretion, etc. We first detected CD69 (early activation marker) and CD25 (late activation marker) expression of T cells in co-culture (PBMC + drug + NY-ESO-1^+^ HLA-A*0201^+^ tumor cell line; E:T = 4:1) after 72 h with FACS. CD69 and CD25 were more up-regulated, and there was an increased IgG-T-TCE-NY concentration (Figure 2a,b). A similar result was observed in the IL-2-detecting experiment in co-culture (Figure 2c). Moreover, we labeled PBMCs with CFSE to see if T cells in co-culture proliferated. After 96 h, T cells proliferated obviously in conditions with a high concentration of IgG-T-TCE-NY (Figure 2d). Overall, the results showed the induced activation of T cells with IgG-T-TCE-NY.

To further check the cytotoxicity of T cells in co-culture, we determined the expression of granzyme B (a cytotoxic protein inducing apoptosis of target cells) in T cells with FACS. During the process of T cells killing target cells, the granzyme B expressed in T cells is released into target cells through direct intercellular contact (immune synapse), and then the granzyme B within target cells activates the apoptosis-related pathway of tumor cells. The up-regulation of granzyme B was observed after 48 h of co-culture, indicating that T cells had an IgG-T-TCE-NY-induced augmented ability to kill tumor cells (Figure 2e). We repeated the experiment and determined the relative quantity of granzyme B in target tumor cells (A375-NY-GFP) using FACS to make clear whether tumor cells were being attacked by T cells. Similar to the result for T cells, the proportion of granzyme B^+^ tumor cells and the quantity of granzyme B in tumor cells were increased with the increased concentration of IgG-T-TCE-NY (Figure 2f), further proving the cytotoxicity of T cells in co-culture.

### 3.3. IgG-T-TCE Redirected T Cells to Specifically Kill Positive Tumor Cells In Vitro

To confirm that IgG-T-TCE-NY can redirect T cells to kill positive tumor cells, the LDH assay and apoptosis assay were conducted for A375 (Figure 3a, left) and A375-NY-GFP (Figure 3a, right), respectively. Both results showed increased tumor-cell killing with the increased concentration of IgG-T-TCE-NY. The killing effect was directly reflected in comparisons of images of co-culture cells, which showed that the number of tumor cells decreased and the morphology of tumor cells changed clearly after adding IgG-T-TCE-NY (Figure 3b).

The specificity of IgG-T-TCE-NY was assessed on two constructed cell lines: K562-NY (NY-ESO-1_157–165_/HLA-A*02:01^+^) and K562-Ctrl (NY-ESO-1_157–165_/HLA-A*02:01^−^). Both cell lines are derived from the same parent cell line K562-A2 (NY-ESO-1^−^ and HLA-A*0201^+^). The result showed obvious killing for K562-NY but not K562-Ctrl, indicating the specificity of IgG-T-TCE-NY for killing positive tumor cells (Figure 3c).

We also investigated in a simple manner the effect of Fc and affinity to CD3 on the cytotoxicity of IgG-T-TCE-NY. The silencing (LALAPG mutation) of Fc decreased the activity of IgG-T-TCE-NY in killing tumor cells, indicating the importance of the Fc function for IgG-T-TCE (Figure 3d). In the experiment on the influence of the anti-CD3 part, the results showed decreased activity for IgG-T-TCE-NY with lower affinity to CD3 (OKT3-LT, 1000-fold lower affinity than OKT3, as shown in the literature [13]) but nearly unchanged activity for IgG-T-TCE-NY with higher affinity to CD3 (UCHT1, 2000-fold higher affinity than OKT3, as reported in [14]), indicating that a certain affinity to CD3 is needed for IgG-T-TCE-NY to sufficiently kill tumor cells but with a threshold (Figure 3e).

### 3.4. IgG-T-TCE-NY Prevented the Growth of NY-ESO-1^+^HLA-A*02:01^+^ Tumors in a Tumor–PBMC Co-Engrafted Mouse Model

To explore the antitumor activity of IgG-T-TCE-NY in vivo, a tumor–PBMC co-engrafted model was established. In detail, NOD-SCID mice were subcutaneously (s.c.) co-injected with A375 and PBMCs at 1:2.5 and then intravenously (i.v.) injected with IgG-T-TCE-NY at a dose of 105 µg/kg thrice every three days (Figure 4a). A no-PBMC group (A375 s.c., PBS i.v.) and a vehicle control group (A375 and PBMC s.c., PBS i.v.) were set as two control groups. Tumor volume was recorded for each mouse at intervals until there was a mouse with a tumor larger than 800 mm^3^. There was no significant change in the weight of mice in each group after treatment (Figure 4b). Curves in Figure 4c,d show slower tumor growth in the IgG-T-TCE-NY group compared with control groups. The same result can be observed in images of tumors taken from mice (Figure 4e).

## 4. Discussion

In this study, we developed an easy-to-obtain TCR-based TCE in an IgG form (IgG-T-TCE) that redirected T cells to specifically and efficiently kill tumor cells with target peptide–MHCs in vitro and to control tumor growth in vivo.

Simply put, the constant (extracellular part) and variable regions of TCR were connected to the constant region of the antibody in one arm, and a pair of cysteines was inserted in the constant region of the TCR to stabilize the TCR. The IgG-T-TCE was obtained through two-step purification (His tag and Flag tag) from the eukaryotic expression system and showed as simultaneous binding targets pMHC and hCD3. Differently from prokaryotic sources, the TCR part in our IgG-T-TCE obtained from the eukaryotic expression system was obviously glycosylated. This N-glycosylation was predicted to be due to multiple glycosylated sites on TCR. It has been clearly shown that glycosylation of TCR on T cells can protect TCR from proteases, increase steric hindrance to decrease nonspecific aggregation, and confer rigidity to TCR to increase interaction with pMHC [16,17]. However, one study showed that T cells with deglycosylated TCR had enhanced avidity to tumor cells and the pMHC tetramer [18], implying that glycosylation may restrict TCR’s binding to pMHC. This needs to be further confirmed because increased TCR clustering could also explain it. We simply compared the binding of our IgG-T-TCE with the pMHC monomer (prokaryotic sources) before and after N-deglycosylation via ELISA. Compared to the ‘before’ group, the ‘after’ group showed no advantage in binding affinity (and even a slight decrease at low concentration) (Figure 1f). One question is whether the glycans on pMHC interact. N-glycosylation inhibition experiments showed that the absence of N-glycans does not modify pMHC’s ability to interact with TCR [16].

The specific lysis of tumor cells mainly depends on the specificity of molecules towards targets. For pMHC, which has a low density on the cell surface (e.g., for NY-ESO-1_157–165_/HLA-A*02:01, 10–50 copies per cell [19]), a monovalent molecule with a high affinity is more effective in enriching molecules on the tumor cell side than a bivalent molecule with a low affinity, which requires a mature screening platform to obtain a specific clone with a high affinity. Compared with the TCR platform [20,21,22], the antibody platform is relatively easy to establish. Some researchers have attempted to develop TCR mimic antibodies to target pMHC [23], but most TCR mimic antibodies exhibited a greater degree of cross-reactivity due to the different recognition modes of TCRs and antibodies towards pMHC [24], which was not conducive to overall specificity. Here, we used a well-identified pMHC (NY-ESO-1_157–165_/HLA-A*02:01) as the cancer-specific antigen and selected a highly specific TCR clone with a very high affinity (1G4-113 clone, Kd at picomole level [12]) to target it. To evaluate the specificity of this IgG-T-TCE, K562-NY (transfected with a plasmid expressing NY-ESO-1_157–165_) and K562-Ctrl (transfected with a plasmid expressing irrelevant peptide) were established as a positive cell line and negative cell line, respectively, based on the same parent cell line K562-A2 we established previously. We compared the apoptosis of two cell lines in the same co-culture condition (PBMC + drug + tumor cell), and the result showed that positive cells demonstrated much more apoptosis than negative cells, indicating this IgG-T-TCE targeted NY-ESO-1_157–165_/HLA-A*02:01 but did not cross-react with the irrelevant one tested here.

We also evaluated the killing efficiency of IgG-T-TCE-NY on cell line A375. The EC_50_ of IgG-T-TCE-NY here was at the picomole level and influenced by multiple factors: with PBMCs, a certain affinity to CD3 was needed for IgG-T-TCE to sufficiently kill tumor cells but with a threshold (Figure 3e); and silence of Fc (IgG1) weakened the ability of IgG-T-TCE (Figure 3d), implying the engagement of FcγR receptor positive immune cells in the elimination of tumor cells. In TCE, Fc acts as a ‘rapier’, adding additional mechanisms of cytotoxicity, like phagocytosis or antibody-dependent cell-mediated cytotoxicity, but also increasing non-specific activation of T cells, leading to a higher risk of cytokine storms [25]. A representative of TCEs that reserve IgG function is Catumaxomab, which was approved in 2009 by the EMA for the treatment of malignant ascites but with a careful administration method for acceptable safety [26]. Besides the dose interruption, additional molecules like kinase inhibitors or cytokine antagonists may reduce cytokine release [27,28,29].

TCEs have shown some clinical efficacy [30]. However, activating T cell receptor signaling with anti-CD3 would not stimulate T cells in a fully activated state, which needs co-stimulatory signaling (CD28, 4-1BB, OX40, etc.) [31]. Recently, a trispecific antibody that interacts with CD38 on tumor cells and CD3 and CD28 on T cells displayed enhanced cytotoxicity to tumor cells [32]. Similar results can be seen for another trispecific antibody that interacts with HER2, CD3, and CD28 [33]. These studies showed that multi-specific TCEs with the synergy from multiple mechanisms may be the next-generation T cell immunotherapy. Benefiting from the good extension ability of IgG, IgG-T-TCE has the potential to be developed into different multi-specific TCEs with high specificity to tumor cells, which is worth exploring.

## 5. Conclusions

In this study, we designed and characterized a novel soluble TCR-based TCE targeting NY-ESO-1_157–165_/HLA-A*02:01-positive tumors. The IgG-T-TCE could be easily obtained from the mammalian cell expression system and showed inspiring antitumor activity in vitro and in vivo. These data demonstrate that IgG-T-TCE is a viable option to target pMHCs for specific tumor killing.

## Figures and Tables

**Figure 1 biomedicines-12-00776-f001:**
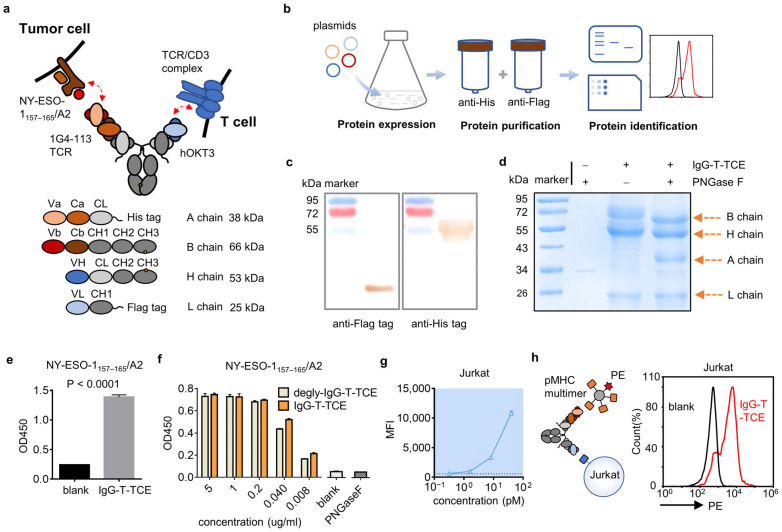
Structure and characterization of IgG-T-TCE. (**a**) Schematic diagram of IgG-T-TCE-NY structure. Red arrows indicate the simultaneous binding of IgG-T-TCE-NY to NY-ESO-1_157–165_/HLA-A*02:01 on tumor cell and human CD3 on T cell. The MW of each chain was estimated using ExPASy. (**b**) Flow diagram of IgG-T-TCE preparation. (**c**) Western blot analysis of IgG-T-TCE-NY. (**d**) Reducing SDS-PAGE analysis of IgG-T-TCE-NY with/without deglycosylation operation. The position of each band matched the estimated size of each chain of IgG-T-TCE-NY after deglycosylation. (**e**) ELISA assay for testing the affinity of IgG-T-TCE-NY to NY-ESO-1_157–165_/HLA-A*02:01. The blank group was incubated with PBS instead of IgG-T-TCE-NY. The t-test of these two groups was performed using GraphPad. *p* value was lower than 0.0001. (**f**) ELISA assay for comparing the affinity of IgG-T-TCE-NY to NY-ESO-1_157–165_/HLA-A*02:01 with/without the deglycosylation operation. PNGase F alone and a blank group incubated with PBS instead of IgG-T-TCE-NY were set as controls. (**g**) Binding ability of IgG-T-TCE-NY to Jurkat cells measured with FACS. (**h**) The simultaneous binding of IgG-T-TCE-NY to NY-ESO-1_157–165_/HLA-A*02:01 and human CD3 was identified with FACS using the sandwich method. The pMHC multimer tagged with PE could bind IgG-T-TCE-NY caught by the hCD3 on Jurkat cells. The blank group was incubated with PBS instead of IgG-T-TCE-NY. Data are means ± SD, *n* = 3.

**Figure 2 biomedicines-12-00776-f002:**
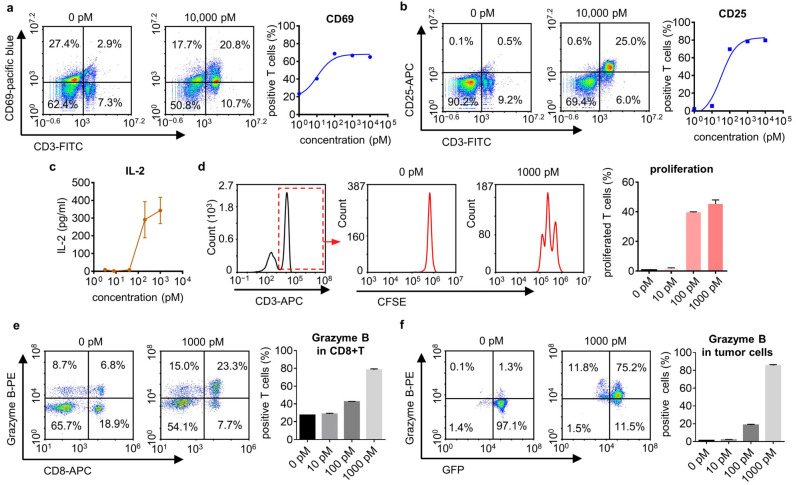
IgG-T-TCE induced T cell activation. (**a**,**b**) Expression of T cell activation markers CD69 and CD25 after 72 h co-culture of PBMCs and positive tumor cells (U266) (E:T, 4:1) in the presence of IgG-T-TCE-NY, measured with FACS. Both CD69 and CD25 expression on T cells increased with the increased concentration of IgG-T-TCE-NY. The colors in the density map represent the cell density at the same location. The cell density increases from blue to red. CD3, FITC; CD69, pacific blue; CD25, APC. (**c**) Analysis of IL-2 secretion after 72 h co-culture (positive tumor cell line: A375). (**d**) Analysis of T cell proliferation after 96 h co-culture (positive tumor cell: A375-NY-GFP). CFSE was used to label PBMC before co-culture. The proportion of proliferated T cells to T cells significantly increased with the increase in IgG-T-TCE-NY concentration. CD3, APC. (**e**) Analysis of granzyme B^+^ CD8^+^ T cells after 48 h co-culture (positive tumor cell line: A375). The proportion of granzyme B^+^ T cells to CD8^+^ T cells significantly increased with the increase in IgG-T-TCE-NY concentration. Granzyme B, PE; CD8, APC. (**f**) Analysis of granzyme B^+^ tumor cells after 48 h co-culture (positive tumor cell line: A375-NY-GFP). The proportion of granzyme B^+^ tumor cells significantly increased with the increase in IgG-T-TCE-NY concentration. Granzyme B, PE. Data are means ± SD, *n* = 3.

**Figure 3 biomedicines-12-00776-f003:**
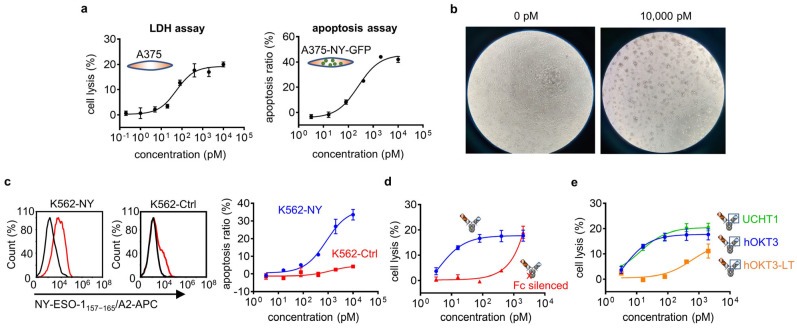
In vitro cytotoxicity assay of IgG-T-TCE. (**a**) Percentage of tumor cell lysis (A375) measured with the LDH assay (left) after 48 h co-culture and tumor cell apoptosis (A375-NY-GFP) detected with FACS after 48 h co-culture (right). (**b**) Representative photomicrographs (40× magnifications) of co-culture cells (positive tumor cell line: A375) in the presence of 0 pM and 10,000 pM IgG-T-TCE-NY after 48 h. (**c**) Relative NY-ESO-1_157–165_/HLA-A*02:01 expression on constructed K562-based cell lines (left) and corresponding tumor cell apoptosis detected with FACS after 48 h co-culture (right). Red line, tumor cells stained with APC-labeled 1G4-113 TCR; black line, blank control. (**d**) Percentage of tumor cell lysis (A375) after 48 h co-culture in the presence of IgG-T-TCE-NY or Fc-silenced IgG-T-TCE-NY measured with the LDH assay. (**e**) Percentage of tumor cell lysis (A375) after 48 h co-culture in the presence of modified IgG-T-TCE-NY with different affinities to human CD3 measured with the LDH assay. Rank order of CD3 affinity: UCHT1 > hOKT3 > hOKT3-LT. Data are means ± SD, *n* = 3.

**Figure 4 biomedicines-12-00776-f004:**
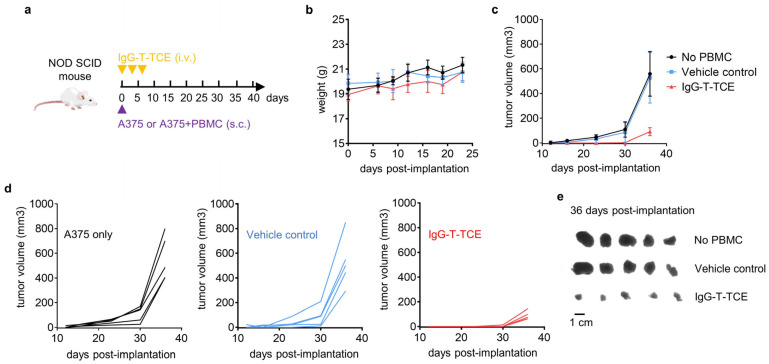
In vivo antitumor activity of IgG-T-TCE. (**a**) Schematic depiction of animal model and experimental design. Female NOD-SCID mice engrafted with A375 melanoma cells (1.5 × 10^6^) and PBMCs (3.75 × 10^6^) were treated with IgG-T-TCE at 0.1 mg/kg according to the schedule depicted by the purple triangle and yellow inverted triangles. As controls, A375 cells were engrafted without or with PBMCs, and mice were dosed with the vehicle (PBS). (**b**,**c**) Average weight and tumor volume change for mice in the three groups. (**d**) Tumor volume change for each mouse in the three groups. (**e**) Digital image of the stripped tumors. Data are means ± SD, *n* = 5.

## Data Availability

The data presented in this study are available upon request from the corresponding author.

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
