# Peer review of "T Cell Receptor-Directed Bispecific T Cell Engager Targeting MHC-Linked NY-ESO-1 for Tumor Immunotherapy"

_biomedicines, 2024, doi:10.3390/biomedicines12040776_

Round 1
Reviewer 1 Report
Comments and Suggestions for Authors
The manuscript „TCR-directed bispecific T cell engager targeting … “ focuses on testing in vitro and in vivo effects of antibody-based bispecific T cell engager, designed for targeting cancer cells carrying tumor-specific antigenic peptide. The authors built their current study on their paper published in Biomedicines 2021, quoted as ref. 10.
The authors have chosen an adequate methodical approach, and the results are sufficiently convincing. I only have a question concerning the granzyme B expression following the co-cultivation of PBMCs with tumor A375 cells. Why was the expression tested also in tumor cells? The data presented here show that the granzyme B was increased in tumor cells similarly as in effector CD8+ T cells (Fig. 2 e, f). What is the significance of this result and why it is not discussed in the Results or Discussion section? This should be explained and clarified.
The manuscript should be corrected from a technical point of view.
In particular, an improvement of the figures is necessary. Some parts of the graphs are hard to discern, such as protein identification (Fig. 1 b, far right, Fig. 1 h, the curve representing positive binding of IgG-T-TCE-NY to Jurkat cells; Fig 3 c). Also, the numbers representing frequency of cell populations in FACS dot plots (Fig 2) should be increased, because they are virtually unreadable both in printed and in digital form.
Comments on the Quality of English Language
The expression „degrees centigrade“ is uncommon and can be changed to „ °C“ (lines 124,125,126,128, 136). Also „homemade pMHC“ is not commonly used: prepared in the laboratory?
Reviewer 2 Report
Comments and Suggestions for Authors
This paper is of high quality and represents a succesful approach to a relatively novel technology of bispecific engagers. The study is complete, and describes all the efforts from the construction of DNA vectros to preclinical in vivo rodent model for assessing antitumor activity.
I can highly recommend publication of this in the present form in Biomedicines.
